# Preparation and Properties of Polyimide/Polysulfonamide/Polyethylene Glycol (PI/PSA/PEG) Hydrophobic Nanofibrous Membranes

**DOI:** 10.3390/ma17164135

**Published:** 2024-08-21

**Authors:** Zijia Wang, Yawen Chang, Siyang Jia, Fujuan Liu

**Affiliations:** National Engineering Laboratory for Modern Silk, College of Textile and Clothing Engineering, Soochow University, 199 Ren-Ai Road, Suzhou 215123, China; 20245215006@stu.suda.edu.cn (Z.W.); 20225215004@stu.suda.edu.cn (Y.C.); 20245215109@stu.suda.edu.cn (S.J.)

**Keywords:** nanofiber membranes, electrospinning, water etching, hydrophobic surface

## Abstract

In this study, polyimide (PI) and polysulfonamide (PSA) were used as base materials, and polyethylene glycol (PEG) was added to successfully prepare PI/PSA/PEG nanofiber membranes through electrospinning technology. Subsequently, water etching was performed on the membranes, utilizing the water solubility of PEG to form the rough wrinkled structure, further enhancing the surface hydrophobicity. The experimental results showed that under the conditions of a spinning voltage of 10 kV, PI/PSA mass fraction of 15 wt.%, and PEG-to-PI/PSA mass ratio of 1/3, the obtained fiber membranes exhibit a uniform morphology (an average diameter of 0.73 µm) and excellent hydrophobicity (the initial water contact angle (WCA) reaching 130.4°). After PEG water etching, the surface of the PI/PSA/PEG hydrophobic membranes formed the rough wrinkled structure, which not only improved their mechanical properties but also further enhanced their hydrophobicity (the initial WCA increasing to 137.9°). Hence, fiber membranes are expected to have broad application prospects in fields such as waterproofing and moisture permeability.

## 1. Introduction

In recent years, with the continuous advancement of technology and the increasing awareness of environmental protection, hydrophobic materials have received extensive attention and research due to their unique self-cleaning ability, excellent waterproof and antibacterial properties [1], as well as their broad applications in areas such as anti-corrosive coatings [2,3], dustproofing [4], anti-icing [5,6], anti-fouling [7], and self-cleaning [8,9,10]. Currently, there are various methods for preparing hydrophobic materials, including etching [11], powder coating [12], deposition [13,14], assembly [15], templating [16], electrospinning [17,18], and 3D printing technology [19]. Among them, electrospinning technology is a simple and effective method for producing nanofiber membranes. Upon passing through a high-voltage electric field, polymer solution or melt forms jets, which are then stretched and refined in the air and ultimately collected on a substrate. These are known as nanofiber membranes. The electrospinning process is influenced by factors such as the spinning voltage, collecting distance, spinning speed, and solution concentration [17,20]. The obtained nanofiber membranes possess significant advantages like a large specific surface area, high porosity, and interconnected porous structure, which enable them to exhibit promising application prospects in air filtration [21,22,23], water purification [24,25,26], oil–water separation [27,28], and other fields [29]. Compared with traditional membrane technology, electrospinning has a lower energy consumption and is, therefore, considered a relatively environmentally friendly technique [30]. Additionally, precise control over the structure and properties of nanofiber membranes can be achieved by adjusting spinning parameters and solution composition.

Moreover, surface modification techniques have become a commonly used method for achieving the hydrophobic modification of materials [31]. This method aims to impart materials with new characteristics, such as hydrophobicity, oleophobicity, acid resistance, and waterproofness, without altering their original performance, thereby enabling them to have a higher value and longer service life in specific environments. Water etching is a technique that utilizes water as an etching medium to remove parts of the material surface under specific conditions, changing its surface properties and endowing specific microstructures. This method can introduce micro–nanoscale rough structures onto the surface of nanofiber membranes, increase the specific surface area and surface energy, and further enhance the waterproof and anti-contamination properties [32].

Polyimide (PI) is a high-performance organic polymer with an imide ring (-CO-NR-CO-) in its main chain. Due to the presence of rigid aromatic heterocyclic structures and strong intermolecular interactions in its molecular backbone, PI exhibits excellent properties such as mechanical, thermal, and radiation resistance; low dielectricity; high insulation; hydrophobicity; and chemical stability [30]. Polysulfonamide (PSA) is a para-aramid series macromolecule with both para- and meta-positions in structure, and its macromolecular chain contains sulfone groups, which has excellent heat resistance, dimensional stability, anti-aging properties, acid and alkali resistance, and radiation resistance, along with low cost [33]. However, PSA’s poor electrostatic resistance, UV resistance, aging resistance, and mechanical properties limit its spinning application, as it is often required to be used in composite form with other fiber materials. Therefore, blending PI, which possesses excellent mechanical strength and tear resistance [34], with PSA for spinning not only combines their superior properties while improving PSA’s poor spinnability [35] but also effectively reduces costs, achieves polymer modification, produces high-performance materials, and further expands their application areas. In addition, polyethylene glycol (PEG) is a well-known high-performance polyether polymer with excellent biocompatibility, water solubility, and non-toxicity, finding widespread applications in the pharmaceutical, hygiene, food, and chemical industries [36].

In this study, the PI/PSA/PEG hydrophobic nanofiber membranes were prepared using high-performance PI and PSA as base materials and PEG as an additive through electrospinning technology. The mass ratios of PEG relative to PI/PSA on the fiber morphologies and diameters were investigated. By analyzing the mechanical properties and hydrophobicity of the nanofiber membranes, the optimal spinning process parameters were determined. Furthermore, by leveraging the water solubility of PEG, a wrinkled and rough surface structure was created on the membranes through water etching. A comparison of the fiber morphology, mechanical properties, and hydrophobicity, before and after etching, revealed that the PI/PSA/PEG nanofiber membranes exhibit improved mechanical properties and enhanced hydrophobicity after water etching. The modified materials will have potential applications in the fields of waterproofing, breathability, moisture permeability, and anti-fouling.

## 2. Materials and Methods

### 2.1. Materials

The polysulfonamide (PSA) staple fibers used in the experiment were purchased from Shanghai Teyarn Co., Ltd. (Shanghai, China). Polyimide (PI, Mw = 40,000 kg/mol) powder was produced by Dongguan Zhanyang Polymer Materials Co., Ltd. (Dongguan, China). N, N-Dimethylacetamide (DMAC, Mw = 87.12 g/mol), Polyethylene glycol (PEG, Mn = 2000 g/mol), methyl orange, prussian blue, and methylene blue were all obtained from Shanghai Aladdin Biochemical Technology Co., Ltd. (Shanghai, China). Tea, milk, and coffee were all self-made in the laboratory.

### 2.2. Preparation of PI/PSA/PEG Nanofiber Membranes

To fabricate the PI/PSA/PEG nanofiber membranes, the following protocol was employed. Initially, PI and PSA were dissolved in DMAC solvent and stirred at room temperature for 5 h to obtain the PI/PSA solution with a concentration of 15 wt.% (where PI:PSA is 7:3). Subsequently, PEG-2000 was incorporated into the solution at varying ratios (1/1, 1/2, 1/3, 1/4, and 1/5) relative to the total PI/PSA mass, and the mixture was further stirred for 12 h to form the desired PI/PSA/PEG spinning solution. The mixture was then transferred into a syringe and electrospinning was performed. The feed rate was set at 1 mL/h, the spinning voltage at 10 kV, and the distance between the spinneret and the plate collection device at 14 cm. Finally, these membranes were dried under vacuum for 2 h to remove residual solvents.

### 2.3. Preparation of Water-Etched PI/PSA/PEG Nanofiber Membranes

In accordance with the scheme detailed in Section 2.2, PI/PSA/PEG nanofiber membranes with different PEG mass ratios to PI/PSA were prepared. Subsequently, the fiber membranes with different PEG mass ratios were subjected to water etching treatment, respectively. Each side was immersed in deionized water for 48 h, followed by drying in an oven at 70 °C for 6 h. The final nanofiber membranes obtained had a unique wrinkled and rough surface morphology, which can be further characterized and applied.

### 2.4. Characterizations

The microstructure of PI/PSA/PEG nanofiber membranes with different PEG addition ratios relative to PI/PSA (1/1, 1/2, 1/3, 1/4, and 1/5), before and after water etching, was examined utilizing scanning electron microscopy (SEM, Regulus 4800, Hitachi Co., Tokyo, Japan). The chemical structures of the membranes with a PEG ratio of 1/3 before and after water etching were analyzed via a Fourier transform infrared spectrometer (FTIR, NICOLET5700 is5, Thermo Fisher Scientific, Madison, WI, USA). The mechanical properties were evaluated using an Instron 5967 universal material testing machine (Instron Corporation, Norwood, MA, USA). The XRD analysis was performed by characterizing the membranes with a PEG ratio of 1/3 before and after water etching using an X-ray diffractometer (XRD, D8 Advance, Bruker, Karlsruhe, Germany) with diffraction angles ranging from 5° to 55°. The dynamic water contact angles (WCAs) were measured by a Theta Flow water droplet angle measurement instrument (Biolin Scientific, Gothenburg, Sweden). The dynamic contact angles reflected the temporal variations in surface contact angles. The measurement duration was determined by different parameters.

The air permeability was tested using the YG461G automatic air permeability meter (Ningfang Instrument, Ningbo, China), with a testing area of 20 cm^2^ and a testing pressure of 100 Pa. The water vapor transmission rate (WVTR) values of the membranes were measured to demonstrate the moisture permeability using test tubes with 50 mL of deionized water. The membranes were sealed and then placed in an incubator at 37 °C and 50% relative humidity for 12 h. The estimation formula for WVTR was read as follows:$$WVTR=\frac{W_{loss}}{S}$$
where *W_loss_* is the daily weight loss of water (kg/day) and *S* is the area of the tube orifice.

## 3. Results and Discussion

### 3.1. Morphology and Structures

Figure 1 illustrates the effect of different PEG mass ratios relative to PI/PSA on the morphology and diameters of PI/PSA/PEG fiber membranes. It can be seen that as the mass ratios of PEG relative to PI/PSA decreased (from 1/1 to 1/5), the average fiber diameters gradually decreased from 0.84 µm to 0.65 µm. Notably, when the PEG mass ratio reached 1/3, the average fiber diameter was about 0.73 µm. At this point, the fibers exhibited excellent uniformity and maintained good continuity with no breaks. Therefore, the PEG-to-PI/PSA mass ratio of 1/3 was an optimal choice for the subsequent preparation of high-quality PI/PSA/PEG fiber membranes.

Figure 2 shows the SEM micrographs and diameter distributions of PI/PSA/PEG fiber membranes after water etching with varying PEG-to-PI/PSA addition ratios. Water etching, a common method of significantly altering the fiber morphology, can dissolve the water-soluble substances, resulting in a general reduction in fiber diameters. Compared with the fiber diameter before water etching (Figure 1), the fiber diameter after water etching decreased evidently (Figure 2). Furthermore, both too-low (1/4 and 1/5) and too-high (1/1) PEG addition ratios led to a decrease in fiber uniformity. Evidently, when the PEG ratio was set to 1/3, the fibers had an average diameter of 0.64 ± 0.03 µm, displaying a uniform and stable morphology.

The magnified SEM images of the fibers after water etching with different PEG addition ratios are shown in Figure 3. It was found that the fiber surfaces became rough and wrinkled, forming a few porous structures after water etching. This may be attributed to the water solubility of PEG, which was partly removed from the nanofiber surface and interior, creating porous structures, thus effectively increasing the fibers’ specific surface area and surface roughness. When the PEG addition ratio was 1/3, the water-etched fibers exhibited noticeably raised rough structures, slight pores, and the most obvious wrinkles.

### 3.2. FTIR Spectrum Analysis

The FTIR spectra profiles of PEG, PI/PSA/PEG, and the etched PI/PSA/PEG membranes are demonstrated in Figure 4. For PI, the strong absorption peak located at 1773 cm^−1^ corresponded to the asymmetric stretching vibration of C=O while the peaks at 1721 cm^−1^ and 1337 cm^−1^ corresponded to the symmetric stretching vibration of C=O and the stretching vibration of C-N in the amide group, respectively. Additionally, the absorption peak at 721 cm^−1^ was attributed to the specific -NH_2_ deformation vibration absorption peak of PSA [35] while the peaks around 1095 cm^−1^ and 2874 cm^−1^ represented the presence of the C-O-C and C-H bond stretching vibration of PEG, respectively. This proved that the PI/PSA/PEG fiber membranes were the composites composed of PI, PSA, and PEG physically combined. The infrared spectrum of the PI/PSA/PEG membranes after water etching still exhibited partial functional group peaks corresponding to PEG, indicating that PEG had not been completely etched away.

### 3.3. XRD Patterns

The XRD patterns of PI/PSA/PEG membranes before and after water etching are illustrated in Figure 5. It can be clearly seen that the PI/PSA/PEG membranes exhibited characteristic diffraction peaks at 2θ of 19.2° and 23.4°, corresponding to the 120 and 032 crystal planes of PEG, respectively [37]. After water etching, the characteristic diffraction peaks of PEG almost disappeared, indicating that the crystal structure of PEG was significantly weakened during the water etching process. This may be due to the infiltration of water molecules into the interior of PEG crystals, weakening the intermolecular interaction force and leading to the destruction of the PEG crystal structure. Therefore, the overall crystal structure of the membranes could be optimized by etching a portion of PEG with water.

### 3.4. Mechanical Properties

Figure 6 shows the stress–strain curves of PI/PSA/PEG fiber membranes with different PEG-to-PI/PSA mass ratios before and after water etching. As the mass ratios decreased from 1/1 to 1/5, the breaking stress of membranes before water etching first decreased and then increased (Figure 6a). This may be due to the formation of large pores between fibers induced by PEG with low mass ratios, resulting in poor mechanics of fiber membranes. However, when the PEG ratio was higher than 1/3 of the critical value, it could effectively inhibit the formation of large pores, making the membrane structure denser and improving the mechanical properties of the membranes. With the decrease in the mass ratios, the breaking stress of the etched membranes decreased first, then increased, and finally decreased (Figure 6b). The breaking stress and breaking strain values of PI/PSA/PEG fiber membranes with different PEG ratios before and after water etching are presented in Table 1. Compared with the unetched membranes, the breaking stress of the water-etched membranes with PEG mass ratios from 1/1 to 1/4 has been improved. This enhancement was attributed to two primary factors: Firstly, water etching removed excess PEG and its dispersed phase, which helped stabilize fiber dimensions and strengthen mechanical properties. In addition, it partially etched PEG, optimizing the crystalline structure for a more orderly molecular arrangement. Consequently, even though fiber membranes initially possessed weaker mechanical properties, post-treatment methods such as water etching can significantly enhance them.

### 3.5. Hydrophobicity Performance

The dynamic water contact angles (WCAs) of PI/PSA/PEG fiber membranes with different PEG ratios are presented in Figure 7. The PI/PSA/PEG fiber membranes prepared with a 1/1 PEG mass ratio exhibited an initial WCA of less than 90°, evidencing hydrophilicity. In contrast, fiber membranes prepared with ratios ranging from 1/2 to 1/5 showed initial WCAs greater than 90° with hydrophobic behavior. As the PEG mass ratios decreased (from 1/1 to 1/5), the initial WCA first increased and then decreased, reaching a maximum of 130.4° (±1.0°) at 1/3. This suggested that at this ratio, the more uniform the fiber diameter distribution, the stronger the surface hydrophobicity. Therefore, the fiber membranes with a PEG mass ratio of 1/3 possessed the highest initial WCA and excellent hydrophobic stability.

Figure 8 displays the dynamic WCA changes of PI/PSA/PEG fiber membranes under different PEG mass ratios after water etching. It can be seen that all membranes processed with water etching exhibited hydrophobicity. With a PEG additive ratio of 1/3, the initial WCA of the membrane was the highest, reaching 137.9° (±0.3°). This was attributed to the decrease in hydrophilicity of the membranes as the PEG mass ratios decreased (from 1/1 to 1/3), resulting in a higher initial WCA. However, when the PEG mass ratio was further reduced, the rough structure on the surface of the membranes became less evident and the water etching effect weakened, thus reducing the hydrophobicity of the surface. In addition, the dynamic WCAs of the etched membranes only decreased by 3.9° within 300 s at a 1/3 PEG mass ratio, which may be due to most of the PEG on the surface of the membranes being dissolved by water. Overall, the hydrophobicity of the membranes after water etching was better than before.

### 3.6. Water Vapor Transmission and Air Permeability

The WVTR and air permeability of PI/PSA/PEG membranes with different PEG mass ratios before and after water etching are presented in Figure 9. Except for the membranes with a PEG ratio of 1/1, the WVTR of the water-etched membranes increased compared to that of the unetched membranes when the PEG ratio was the same (Figure 9a). Particularly, the maximum WVTR of 4.45 ± 0.35 kg/(m^2^·day) was observed for the etched membrane with a PEG ratio of 1/5. This may be due to the fact that the etched membranes were mainly composed of hydrophobic PI and PSA, making it difficult for water vapor to directly penetrate the fiber membranes. The etching effect, on the other hand, might enlarge the pore size between the fibers or increase the number of pores, thus providing more channels for water vapor. Notably, at lower PEG ratios (from 1/2 to 1/5), it was conducive to the formation of more pore structures, and the increase in WVTR after water etching was more pronounced, indicating that water etching could improve the moisture permeability of the membranes.

As shown in Figure 9b, the air permeability of the etched membranes exhibited a trend in first increasing and then decreasing. When the PEG ratio was 1/1, the air permeability of the etched membranes was lower than that of the unetched membranes. This may be because the adhesion between the fibers decreased after the PEG was partially etched by water, and the larger pore size became smaller (Figure 1a and Figure 2a), resulting in a decrease in the air permeability of the etched membranes. However, when the ratio of PEG changed from 1/2 to 1/5, the air permeability of the etched membranes generally increased compared to that of the unetched membranes. The reason for this may be that the water etching caused the fiber diameter to become thinner and the pores between the fibers larger, thereby improving the air permeability. Especially, the air permeability of the etched PI/PSA/PEG membranes reached the highest level (46.16 ± 1.83 mm/s) when the PEG ratio was 1/3.

### 3.7. Waterproofing, Moisture Vapor Transmission, Air Permeability, and Anti-Soiling Properties

By observing the droplet morphology after dripping different liquids onto the surface of the fiber membranes (Figure 10), it was found that the membranes before water etching were completely wet within 1 min. However, the droplets can maintain their near-spherical shape for a long time on the surface of the membrane after water etching, once again indicating that the etched PI/PSA/PEG fiber membranes have good hydrophobic properties.

Figure 11 demonstrates the waterproof performance, breathability, moisture permeability, and anti-fouling performance of etched PI/PSA/PEG fiber membranes. From Figure 11a, we can observe that the membrane could withstand 250 mL of water for a period of time, during which no water leaked out and the membrane remained intact. During the process of increasing from room temperature (22 °C) to 100 °C, the color-changing silica gel gradually changed from the original blue to a light pink (Figure 11b), indicating that the etched PI/PSA/PEG fiber membrane allowed moisture penetration and had moisture permeability. As N_2_ was introduced, the etched membrane allowed N_2_ to pass through while preventing water (Figure 11c). As shown in Figure 11d, when rinsed with deionized water, the soil on the surface of the etched membrane was quickly removed, leaving the etched fiber membrane clean and intact. To some extent, this confirmed that the etched PI/PSA/PEG fiber membranes had good anti-fouling properties.

## 4. Conclusions

In summary, PI/PSA/PEG nanofibrous membranes with surface-roughened wrinkled structures have been successfully prepared through electrospinning combined with water etching. Before water etching, the PI/PSA/PEG fiber membrane composites were physically assembled by PI, PSA, and PEG, and it is worth noting that PEG was not completely etched away during the water etching process. Under the optimal spinning conditions of a spinning voltage of 10 kV, a PI/PSA mass fraction of 15 wt.%, and a PEG mass ratio of 1/3 relative to PI/PSA, the etched PI/PSA/PEG fiber membranes exhibited a uniform and stable morphology with an average diameter of 0.64 ± 0.3 µm, and the mechanical properties could be improved by water etching. Additionally, the initial WCA of the etched membrane reached 137.9°, indicating an improvement in hydrophobicity. The results revealed that the etched PI/PSA/PEG membranes with a rough and wrinkled surface have demonstrated endless potential in the fields of waterproofing, moisture permeability, breathability, and anti-fouling.

## Figures and Tables

**Figure 1 materials-17-04135-f001:**
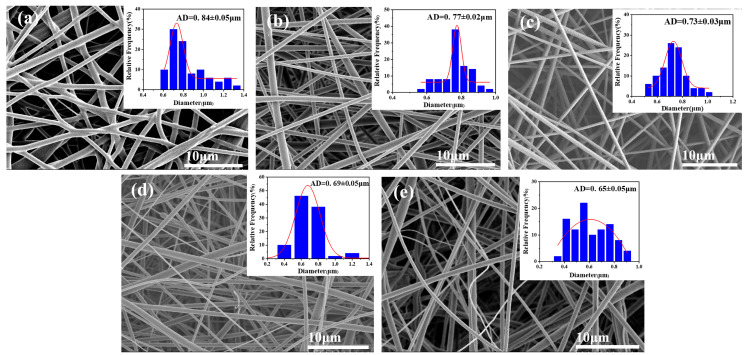
SEM micrographs and diameter distribution histograms of PI/PSA/PEG fiber membranes with different PEG-to-PI/PSA mass ratios of 1/1 (**a**); 1/2 (**b**); 1/3 (**c**); 1/4 (**d**); and 1/5 (**e**).

**Figure 2 materials-17-04135-f002:**
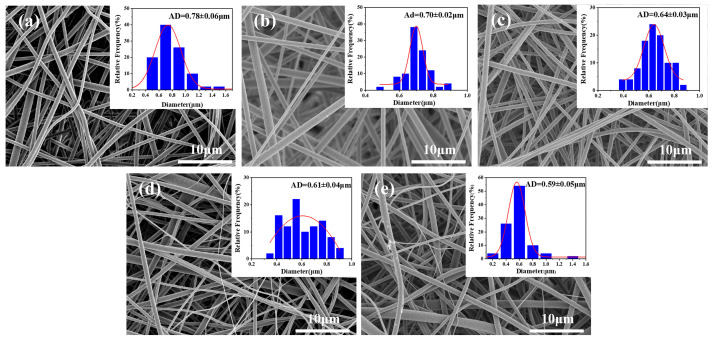
SEM micrographs and diameter distribution histograms of PI/PSA/PEG fiber membranes after water etching with different PEG-to-PI/PSA mass ratios of 1/1 (**a**); 1/2 (**b**); 1/3 (**c**); 1/4 (**d**); and 1/5 (**e**).

**Figure 3 materials-17-04135-f003:**
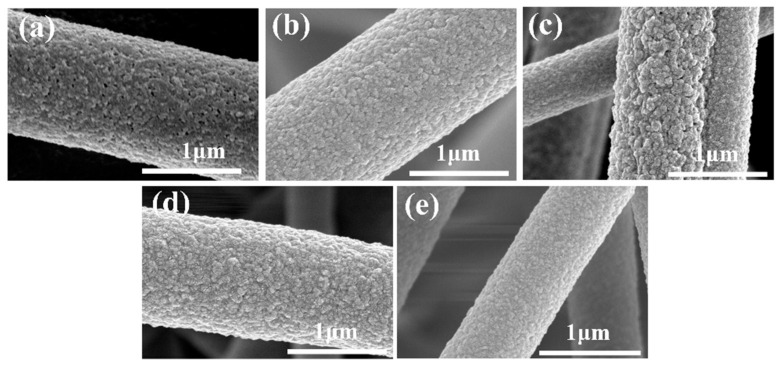
Magnified SEM micrographs of etched PI/PSA/PEG fiber membranes with different PEG-to-PI/PSA mass ratios of 1/1 (**a**); 1/2 (**b**); 1/3 (**c**); 1/4 (**d**); and 1/5 (**e**).

**Figure 4 materials-17-04135-f004:**
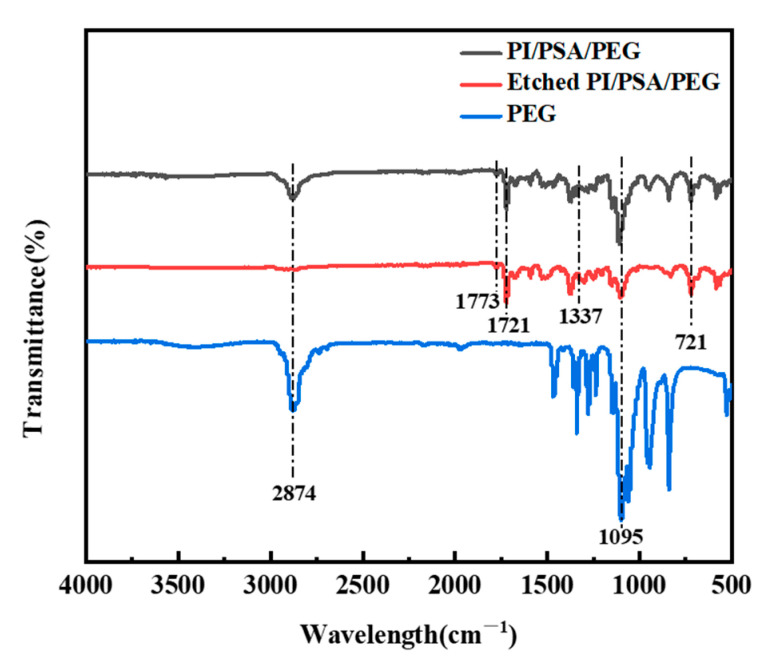
FTIR spectrum profiles of PEG, PI/PSA/PEG, and etched PI/PSA/PEG fiber membranes.

**Figure 5 materials-17-04135-f005:**
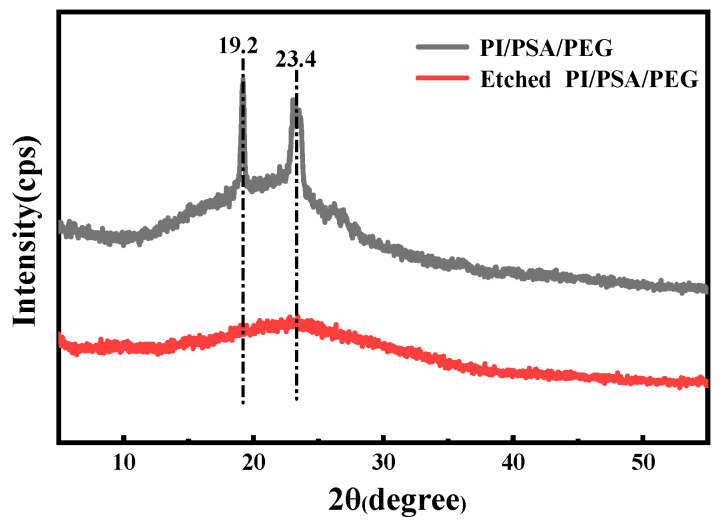
XRD spectra of PI/PSA/PEG and etched PI/PSA/PEG fiber membranes.

**Figure 6 materials-17-04135-f006:**
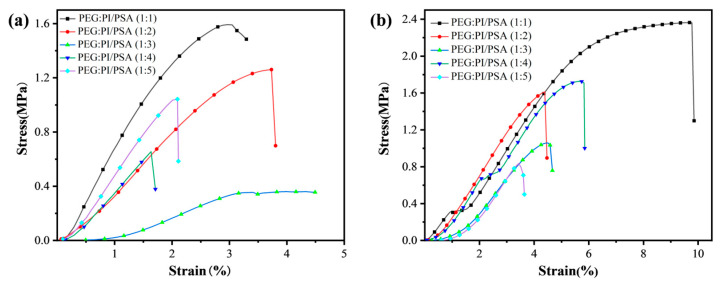
Stress–strain curves of PI/PSA/PEG fiber membranes with different PEG-to-PI/PSA mass ratios before (**a**) and after (**b**) water etching.

**Figure 7 materials-17-04135-f007:**
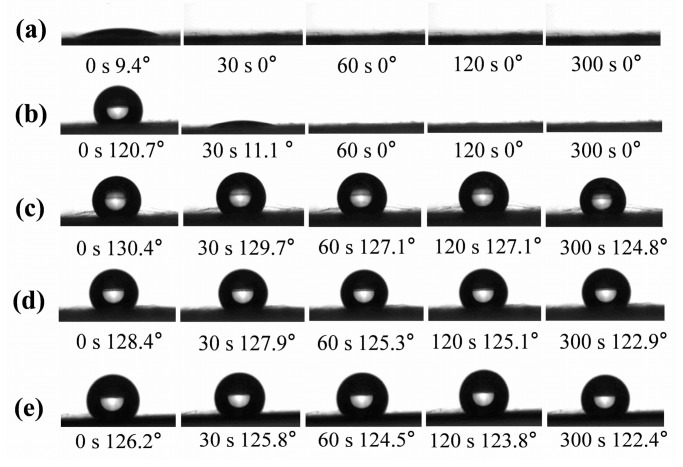
Dynamic WCAs of PI/PSA/PEG nanofiber membranes with different PEG-to-PI/PSA mass ratios of 1/1 (**a**); 1/2 (**b**); 1/3 (**c**); 1/4 (**d**); and 1/5 (**e**).

**Figure 8 materials-17-04135-f008:**
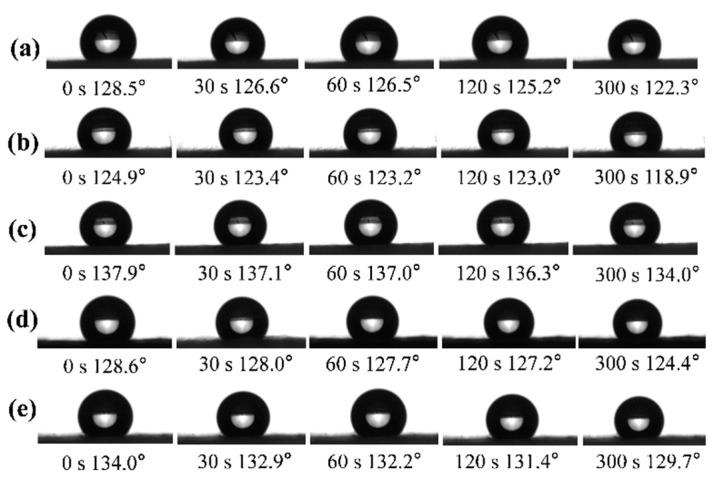
Dynamic WCAs of PI/PSA/PEG nanofiber membranes after water etching with different PEG-to-PI/PSA mass ratios of 1/1 (**a**); 1/2 (**b**); 1/3 (**c**); 1/4 (**d**); and 1/5 (**e**).

**Figure 9 materials-17-04135-f009:**
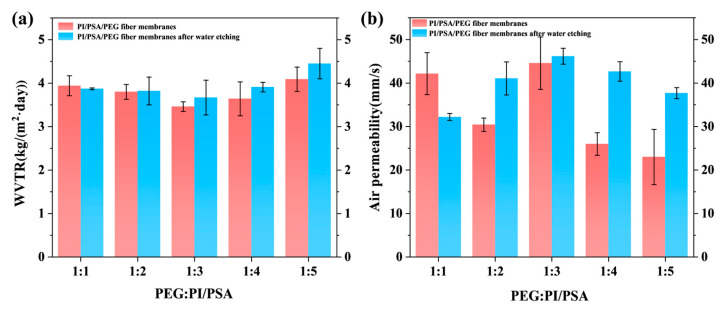
WVTR (**a**) and air permeability (**b**) of PI/PSA/PEG membranes with different PEG-to-PI/PSA mass ratios before and after water etching.

**Figure 10 materials-17-04135-f010:**
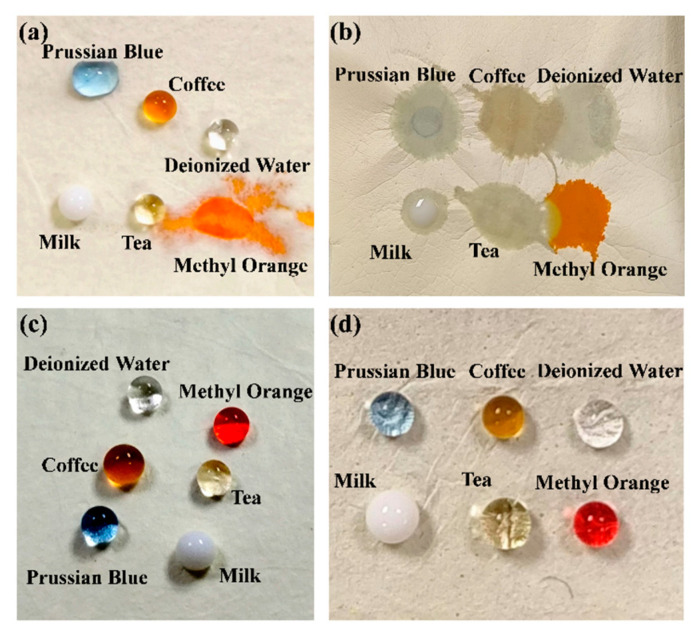
Phenomena of different droplets on the surface of PI/PSA/PEG membranes (**a**); PI/PSA/PEG membranes after 1 min (**b**); etched PI/PSA/PEG membranes (**c**); and etched PI/PSA/PEG membranes after 10 min (**d**).

**Figure 11 materials-17-04135-f011:**
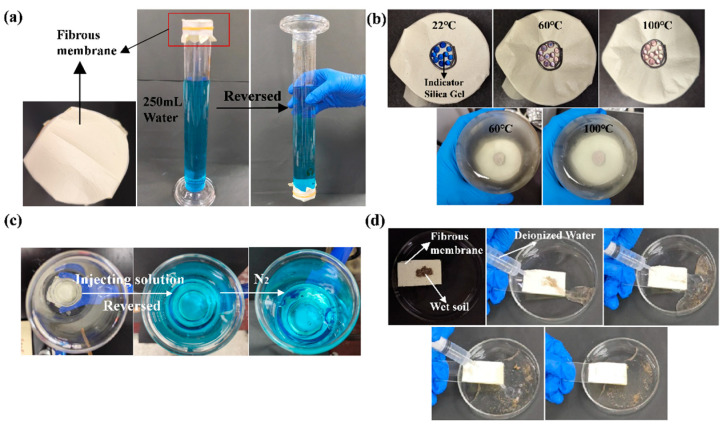
Performance tests on etched PI/PSA/PEG nanofiber membrane: waterproofness (**a**); moisture permeability (**b**); breathability (**c**); and anti-fouling performance (**d**).

**Table 1 materials-17-04135-t001:** Breaking stress and breaking strain of PI/PSA/PEG fiber membranes with different PEG-to-PI/PSA mass ratios before and after water etching.

Fiber Membranes	PEG: PI/PSA	Breaking Stress (MPa)	Breaking Strain (%)
PI/PSA/PEG fiber membranes before water etching	1:1	1.61 ± 0.22	4.42 ± 1.59
	1:2	1.22 ± 0.29	3.50 ± 0.50
	1:3	0.51 ± 0.09	2.15 ± 0.71
	1:4	0.64 ± 0.18	1.53 ± 0.20
	1:5	1.04 ± 0.04	1.93 ± 0.02
PI/PSA/PEG fiber membranes after water etching	1:1	2.37 ± 0.32	9.88 ± 2.74
	1:2	1.63 ± 0.11	4.67 ± 0.83
	1:3	1.09 ± 0.06	3.79 ± 0.41
	1:4	1.61 ± 0.31	5.07 ± 0.70
	1:5	0.82 ± 0.18	3.10 ± 0.44

## Data Availability

The original contributions presented in the study are included in the article, further inquiries can be directed to the corresponding author.

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
