# Peer review of "Preparation and Properties of Polyimide/Polysulfonamide/Polyethylene Glycol (PI/PSA/PEG) Hydrophobic Nanofibrous Membranes"

_materials, 2024, doi:10.3390/ma17164135_

Round 1
Reviewer 1 Report
Comments and Suggestions for Authors
Recently, more and more works devoted to the production of nanofibers have appeared. This is not surprising since the easiest way to achieve the unique properties of a fiber is to reduce its diameter.
There are numerous methods for producing fibers with small diameters. Among them, electrospinning occupies a serious position. This manuscript suggests using a ternary system based on Polyimide/Polysulfonamide/Polyethylene Glycol (PI/PSA/PEG).
The resulting fibers are hydrophobic and it is proposed to obtain membranes from them. I would like to immediately note that fibers, by their nature, often have complex shapes and geometries, so “with Rugged Wrinkled Surfaces” can be removed from the title of the manuscript. This will simplify the title and not greatly affect its meaning.
The abstract contains the main idea and results of the work.
It is possible to remove "waterproof" from the list of keywords or replace this keyword.
58. How are anti-contamination properties improved?
The degree of polymerization or molecular weight should be provided in the description section of the polymers used. The methods used need to be described in more detail.
Section 2.3. "Preparation of Water-etched PI/PSA/PEG Nanofiber Membranes" definitely needs to be expanded. It is not clear to me now what actions were performed with the samples.
The data presented in Figure 5 confuses me a little. Why are the strength values ​​so low? And how was the strength measured for such thin fibers?
The course of the curve in Figure 5b for the 1:1 sample is not entirely clear. The authors write about the structure, but do not provide X-ray diffraction data, etc. Why?
Conclusions briefly present the results obtained in the work.
Author Response
Comments1: The resulting fibers are hydrophobic and it is proposed to obtain membranes from them. I would like to immediately note that fibers, by their nature, often have complex shapes and geometries, so “with Rugged Wrinkled Surfaces” can be removed from the title of the manuscript. This will simplify the title and not greatly affect its meaning.
Reply1: Thanks for the good suggestion.
We have removed the phrase "with Rugged Wrinkled Surfaces" from the title.
Comments2: The abstract contains the main idea and results of the work.
Reply2: Thank you for your valuable comment.
In fact, the abstract clearly outlines the main idea of this research, which is to fabricate PI/PSA/PEG membranes through a combination of electrospinning and water etching. The study also summarizes the main results: the membranes at a PEG ratio of 1/3 have uniform morphology, enhanced hydrophobicity and mechanical properties, which have potential applications in water repellency and moisture permeability.
Comments3: It is possible to remove "waterproof" from the list of keywords or replace this keyword.
Reply3: Thanks for the good suggestion.
The "waterproof" from the list of keywords upon further consideration has been removed.
Comments4: How are anti-contamination properties improved?
Reply4: Thanks for the helpful comment.
The water etching technique uses water as the etching medium to remove certain parts of the material surface with the help of physical or chemical methods by adjusting the etching conditions, ,which can introduce micro-nano scale rough structures on the surface of nanofiber membranes. It is worth noting that hydrophobic surfaces with micro-nano structures have better antifouling properties. To ensure the rigor of our discussion, a supporting reference 32 has been provided. For details, please refer to Page2, Line 58, highlighted in red.
Comments5: The degree of polymerization or molecular weight should be provided in the description section of the polymers used. The methods used need to be described in more detail. Section 2.3. "Preparation of Water-etched PI/PSA/PEG Nanofiber Membranes" definitely needs to be expanded. It is not clear to me now what actions were performed with the samples.
Reply5: Thanks for your careful reading.
The molecular weights of the polymers have been added in Lines 92-94 of Section 2.1 on Page 2, marked in red.
For ease of understanding, we have provided a more detailed description of etched membranes in Section 2.3. The specific content is marked in red on Lines 108-109 of the Section 2.3 on Page 3.
Comments6: The data presented in Figure 5 confuses me a little. Why are the strength values so low? And how was the strength measured for such thin fibers?
Reply6: Thanks for your careful reading.
As is well known, the diameters of electrospun fibers are generally in the micro-nano range, making it very difficult to measure the mechanical properties of individual fibers. In this work, we focused on measuring the mechanical properties of nanofiber membranes, which was more feasible and practical. In the mechanical performance results, it has been indicated that the testing was conducted on the fiber membranes, not on individual fibers. Please see Page3, Lines 120-121 for specific details.
The main reasons for the low mechanical strength of electrospun nanofiber membranes are as follows. Firstly, the unstable "whipping" phenomenon during the electrospinning leads to the random orientation of the polymer chains and crystal regions, which weakens the synergistic effect between the fibers. Secondly, the high porosity structure increases the air gap inside the fiber membranes, thereby reducing the density and mechanical strength. Finally, the inter-fiber interaction force in electrospun nanofiber membranes is weak and lacks effective connection points. Some related studies also support these viewpoints. For example, Yin et al.[1] prepared ultrafine sulfonated polyether sulfone (SPES) membranes by two-nozzle electrospinning, with a maximum tensile strength of 2.05 ± 0.18 MPa, mainly due to their ultrafine fiber diameter (62 ± 16 nm) and high porosity. Augustine et al.[2] fabricated polycaprolactone/ZnO nanocomposite membranes by electrospinning. When ZnO content was greater than 1 wt.%, it would cause a sharp decrease in the crystallinity of the polymer and reduced the mechanical properties of the composite membranes. The tensile strength reached a maximum value of 1.60 ± 0.23 MPa when the ZnO content was 1 wt.%.
References
1 Yin, X.; Zhang, Z.J.; Ma, H.Y.; Venkateswaran, S.; Hsiao, B.S. Ultra-fine electrospun nanofibrous membranes for multicomponent wastewater treatment: Filtration and adsorption. Separation and Purification Technology 2020, 242, 116794. https://doi.org/10.1016/j.seppur.2020.116794
2 Augustine, R.; Malik, H.N.; Singhal, D.K.; Mukherjee, A.; Malakar, D.; Kalarikkal, N.; Thomas, S. Electrospun polycaprolactone/ZnO nanocomposite membranes as biomaterials with antibacterial and cell adhesion properties. Journal of Polymer Research 2014, 21, 347. https://doi.org/10.1016/10.1007/s10965-013-0347-6
In addition, no reinforcing materials were added to the electrospun membranes in this study, which was the reason why the mechanical strength of electrospun fiber membranes was generally low.
Comments7: The course of the curve in Figure 5b for the 1:1 sample is not entirely clear. The authors write about the structure, but do not provide X-ray diffraction data, etc. Why?
Reply7: Thanks for the good suggestion.
As shown in the mechanical performance curves of Section 3.4, the etched membranes with a PEG mass ratio of 1/1 exhibited an increase in the breaking stress and strain compared to the unetched membranes. Table 1 also displayed the specific values of breaking stress and strain.
In order to gain a clearer understanding of the chemical composition and crystal structure of the fibers, we conducted XRD experiments and analysis, as shown in Figure 5. For detailed modifications, please refer to the marked content in red in Section 2.4 (Lines 121-123 on Page 3), Section 3.3 (on Pages 5-6) and reference 37 .
Comments8: Conclusions briefly present the results obtained in the work.
Reply8: Thank you for the helpful comment.
The conclusions of this work presented that the successfully prepared etched PI/PSA/PEG membranes had good morphology, enhanced hydrophobicity and mechanical properties at a PEG ratio of 1/3, with great potential in the fields of waterproofing, moisture permeability, and antifouling.
Reviewer 2 Report
Comments and Suggestions for Authors
The authors present a manuscript studying the preparation and properties of Polyimide/Polysulfonamide/Polyethylene Glycol (PI/PSA/PEG) hydrophobic membranes with wrinkled surfaces.
The manuscript is interesting but, in my opinion, needs major revision before it can be considered for publication.
There are some characterizations which were only performed for a selected etched sample that was not previously identificated.
Moreover, in my opinion, the authors must include FTIR analyses for the whole set of samples, with and without etching treatment.
In addition, the waterproofing, moisture vapor transmission, air permeability, and anti-soiling properties must be also performed for the whole set of samples.
Otherwise the article only shows partial information of the nanofiber membranes and the effect of etching.
There are some other minor issues that need to be revised.
For instance, the terms DMAC (page 3, line 99) and WCA (page 3, line 120) were not previously defined in the manuscript.
Author Response
Comments1: There are some characterizations which were only performed for a selected etched sample that was not previously identificated.
Reply1: Thanks for your careful reading.
The proportions of samples selected for characterization was supplemented in the ‘Characterizations’ of Section 2.4. The Specific revisions can be found in red text in Lines 115-119.
Comments2: Moreover, in my opinion, the authors must include FTIR analyses for the whole set of samples, with and without etching treatment.
Reply2: Thanks for the comment.
In this work, the PI/PSA/PEG nanofiber membranes were simply physically assembled from PI, PSA, and PEG without any chemical changes between the components. Samples with different PEG ratios had similar functional groups, and their FTIR spectra were expected to show very similar characteristic peaks. Thus, only fiber membranes with a PEG ratio of 1/3 (better morphology) were subjected to FTIR testing before and after water etching.
Comments3: In addition, the waterproofing, moisture vapor transmission, air permeability, and anti-soiling properties must be also performed for the whole set of samples. Otherwise the article only shows partial information of the nanofiber membranes and the effect of etching.
Reply3: Thanks for the good suggestion.
The moisture permeability and air permeability tests of the whole set of samples have been supplemented, and the results have been analyzed. The specific modifications can be found in the red marking of Section 2.4 (Page 3, Lines 128-135) and Section 3.6 (Pages 8-9, Lines 254-282). However, the waterproof performance of fiber membranes is quantitatively evaluated by applying hydrostatic pressure, while the mechanical properties of fiber membranes are relatively weak and cannot withstand the whole process of hydrostatic pressure testing, making it difficult to obtain specific waterproof performance data. In addition, there is currently no fully standardized method for antifouling performance testing. Our main goal is to propose the possibility of using etched PI/PSA/PEG fiber membranes for anti-soiling performance testing in the future through this work.
Comments4: There are some other minor issues that need to be revised. For instance, the terms DMAC (page 3, line 99) and WCA (page 3, line 120) were not previously defined in the manuscript.
Reply4: Thanks for the careful reading.
We have carefully checked the manuscript and found that DMAC was defined in Section 2.1, Line 93. Regarding WCA, we have supplemented the definition in Section 2.4. In addition, the necessary changes have been made to the manuscript, please refer to Line 124 on Page 3 ,marked in red.
Round 2
Reviewer 1 Report
Comments and Suggestions for Authors
21. Angle correct to angle.
193-195. Do I understand correctly that the authors mean the swelling of the sample here? What is the moisture content of the sample - etched PI/PSA/PEG fiber membranes?
316. good morphology - not a good phrase.
I recommend using one of the word variants "antifouling" or "anti-fouling"
Comments on the Quality of English Language316. good morphology - not a good phrase.
I recommend using one of the word variants "antifouling" or "anti-fouling"
Author Response
Comments1: Angle correct to angle.
Reply1: Thanks for the good suggestion.
"Angle" in Line 21 has been changed to "angle" as requested and marked in yellow.. Other similar issues in the manuscript have also been modified and marked.
Comments2: 193-195. Do I understand correctly that the authors mean the swelling of the sample here? What is the moisture content of the sample - etched PI/PSA/PEG fiber membranes?
Reply2: Thank you for your valuable comment.
Lines 193-195 in the manuscript explored the effect of water etching on the crystal structure of PEG, which was related to the water solubility of PEG rather than the swelling of the samples due to water absorption. Since PEG is a water-soluble polymer with a large number of ethoxylates in its molecular structure, it can form numerous hydrogen bonds (H-bonds) with H3O in the water molecules[1] allowing the water molecules to penetrate into PEG crystals more easily by overcoming the energy barriers within the crystals. In addition, the flexibility and mobility of the PEG molecular chains also contribute to the diffusion of water molecules between them. With the continuous penetration of water molecules, the existing H-bonding network inside the PEG crystals is gradually replaced by newly formed PEG-water H-bonds, leading to a weakened crystal structure.
It should be noted that the water-etched PI/PSA/PEG fiber membranes had been dried in an oven, so there was almost no moisture content inside. For details, please refer to Lines 109-111on Page 3.
References
1 Cao, N.P.; Zhao, Y.H.; Chen, H.B.; Huang, J.Y.; Yu, M.; Bao, Y.; Wang, D.P.; Cui, S.X. Poly(ethylene glycol) Becomes a Supra-Polyelectrolyte by Capturing Hydronium Ions in Water. Macromolecules 2022, 55, 4656-4664. https://doi.org/10.1016/10.1021/acs.macromol.2c00014
Comments3: 316. good morphology - not a good phrase.
Reply3: Thanks for your careful reading.
We have replaced "the good morphology" with "a uniform and stable morphology" in Lines 316-317, marked the changes in yellow.
Comments4: I recommend using one of the word variants "antifouling" or "anti-fouling".
Reply4: Thanks for your helpful recommendation.
We have carefully checked the manuscript and consistently used the "anti-fouling" variant. For details, please refer to Section 1 (Page 1, Line 33) and Section 3.7 (Page 10, Line 305), highlighted in yellow.
Reviewer 2 Report
Comments and Suggestions for Authors
Authors have addressed my major concerns about the first submission and, in my opinion, the revised manuscript worth for publication.
Author Response
Comments: Authors have addressed my major concerns about the first submission and, in my opinion, the revised manuscript worth for publication.
Reply: Thanks for your insightful comments and support.
We appreciate your positive assessment of the revised version, and we are delighted to hear that you consider it worthy of publication. Your suggestions have been invaluable in improving the quality of the work.